# Caspase-Linked Programmed Cell Death in Prostate Cancer: From Apoptosis, Necroptosis, and Pyroptosis to PANoptosis

**DOI:** 10.3390/biom13121715

**Published:** 2023-11-28

**Authors:** Minggang Zhu, Di Liu, Guoqiang Liu, Mingrui Zhang, Feng Pan

**Affiliations:** 1Department of Urology, Union Hospital, Tongji Medical College, Huazhong University of Science and Technology, Wuhan 430022, China; zhuminggang@hust.edu.cn (M.Z.); didiana@126.com (D.L.); m202275972@hust.edu.cn (M.Z.); 2Urology Department of Guangzhou First People’s Hospital, Guangzhou 510000, China; m202075932@hust.edu.cn

**Keywords:** prostate cancer, apoptosis, necroptosis, pyroptosis, PANoptosis

## Abstract

Prostate cancer (PCa) is a complex disease and the cause of one of the highest cancer-related mortalities in men worldwide. Annually, more than 1.2 million new cases are diagnosed globally, accounting for 7% of newly diagnosed cancers in men. Programmed cell death (PCD) plays an essential role in removing infected, functionally dispensable, or potentially neoplastic cells. Apoptosis is the canonical form of PCD with no inflammatory responses elicited, and the close relationship between apoptosis and PCa has been well studied. Necroptosis and pyroptosis are two lytic forms of PCD that result in the release of intracellular contents, which induce inflammatory responses. An increasing number of studies have confirmed that necroptosis and pyroptosis are also closely related to the occurrence and progression of PCa. Recently, a novel form of PCD named PANoptosis, which is a combination of apoptosis, necroptosis, and pyroptosis, revealed the attached connection among them and may be a promising target for PCa. Apoptosis, necroptosis, pyroptosis, and PANoptosis are good examples to better understand the mechanism underlying PCD in PCa. This review aims to summarize the emerging roles and therapeutic potential of apoptosis, necroptosis, pyroptosis, and PANoptosis in PCa.

## 1. Introduction

Prostate cancer (PCa) is the most common type of nonskin malignancy and one of the leading causes of cancer-related death in men [1,2]. Owing to PCa heterogeneity, patients in different clinical states benefit from different treatments [3,4]. Surgery and radiation therapy are common forms of treatment for localized PCa [5]. Considering that 25~35% of PCa patients will relapse and develop advanced PCa, androgen deprivation therapy (ADT) is recognized as the cornerstone therapy for recurrent PCa. However, almost all patients will develop resistance to androgen deprivation therapy and inevitably relapse into the hormone-independent stage, that is, castration-resistant prostate cancer (CRPC), within a few years [2,6,7]. In this case, a considerable number of studies have focused on the mechanism and therapeutic potential of inhibiting the androgen receptor (AR) pathway. Nevertheless, therapies targeting the AR pathway have gradually been shown to be resistant [8]. PCD has been a hot spot in the spotlight in recent years. Mechanisms underlying apoptosis, necroptosis, pyroptosis, and PANoptosis in PCa are perfect subjects with which to study the role of PCD in PCa.

Cell death is as important as cell proliferation in the development and homeostasis of multicellular organisms [9,10]. As a crucial member of cell death, PCD plays an indispensable role in maintaining biological balance in cells and tissues and responding to infection, tumors, and some other pathologies [11,12]. Apoptosis, pyroptosis, and necroptosis are three extensively studied and understood forms of PCD. As the earliest form of PCD to be discovered, apoptosis is a nonlytic PCD with an integral cellular membrane and is considered immunologically silent [9]. Necrosis is a lytic and inflammatory form of unregulated and accidental cell death. However, studies have revealed that some subtypes of necrosis are driven by specific molecules, and these molecules determine the cell death modality named programmed necrosis or necroptosis [13]. Necroptotic cells undergo swelling and membrane rupture, which causes the release of intracellular damage-associated molecular patterns (DAMPs) into the extracellular microenvironment [14,15]. Pyroptosis is characterized by a series of events initiated by rapid caspase-dependent plasma membrane rupture and the release of inflammation-inducing intracellular contents [16,17,18]. Apoptosis, necroptosis, and pyroptosis are mediated by specific pathways and molecules that eliminate damage factors. Furthermore, these pathways can interact and cooperate with each other [19,20]. Notably, PANoptosis, a novel complex form of PCD, is a combination of the three aforementioned forms of cell death and is triggered in people with infectious diseases and cancer [21,22]. PANoptosis reveals the importance of these three forms of death and may contribute to anticancer drug development. Furthermore, therapy targeting the mechanism underlying apoptosis has led to brilliant results in patients with PCa, which also shows the importance of continuing to develop drugs that target necroptosis and pyroptosis pathways. In this review, the mechanisms of several characteristic forms of PCD are elaborated. The emerging representative treatment and targets for the four forms of PCD are summarized.

## 2. The Mechanisms of Apoptosis, Necroptosis, and Pyroptosis and Their Connection

Specifically, we state that apoptosis, necroptosis, and pyroptosis are caspase-linked and have specific molecular mechanisms. However, PANoptosis, being a composite form of these three types of cell death, still lacks a clear understanding of its specific molecular mechanisms in current research. Hence, we will focus on introducing the molecular mechanisms of apoptosis, necroptosis, pyroptosis, and their interconnections, while excluding PANoptosis.

### 2.1. Mechanisms of Apoptosis

First proposed by Kerr et al. [23] in 1972, apoptosis is characterized by the shrinkage and condensation of cells, the crumbling of the nuclear envelope, the condensation of chromatin with fragmentation of DNA, and the formation of small apoptotic vesicles known as ApoBDs [24]. Apoptotic cells do not release cellular contents into their surroundings and activate an inflammatory response, which makes apoptosis a mild type of PCD [25].

Two major pathways contribute to cell apoptosis: the intrinsic pathway involving B-cell lymphoma-2 (BCL-2) family proteins and the extrinsic pathway mediated by death receptor (DR) ligands [19] (Figure 1). The extrinsic pathway is initiated by the activation of apoptotic death receptors, including tumor necrosis factor (TNF) receptor 1 (TNFR1), Fas, and TNF-related apoptosis-inducing ligand (TRAIL) receptors (DR4 and DR5) when bound to their ligands (TNFα, FasL, and TRAIL) [26,27,28]. Then, adaptor proteins are recruited, namely TNF receptor-associated death domain (TRADD) and Fas-associated death domain protein (FADD), which activate the downstream interactor procaspase-8/10 and form the death-inducing signaling complex (DISC) [29]. Activated caspase-8 initiates the execution phase of apoptosis by cleaving the downstream effector caspase-3 or caspase-7 [30]. Inhibitors of apoptosis (IAP) proteins such as XIAP and IAP1/2 can bind to and inhibit the activation of caspase-3/9 [31]. In the intrinsic pathway, stimuli such as toxicity-inducing substances or DNA damage dysregulate intracellular homeostasis and cause mitochondrial outer membrane permeabilization (MOMP), which leads to the release of cytochrome c into the cytosol. Cytochrome-c, regulated by the Bcl-2 family, recruits procaspase-9 by binding to apoptotic protease-activating factor-1 (APAF-1), which triggers the formation of apoptosomes [32,33]. Through the apoptosome, procaspase-9 is cleaved to generate activated caspase-9, which then activates the effector caspase-3 [34]. The Bcl-2 family plays an indispensable role in regulating mitochondrion-related apoptosis and can be divided into proapoptotic proteins (such as Bax and Bak) and antiapoptotic proteins (such as Bcl-2 and Bcl-XL) [35]. Another important proapoptotic protein, namely BH3-only proteins, shares a homologous BH3 region, through which they bind and directly activate Bax and/or Bak-1 or inhibit antiapoptotic proteins to mediate apoptosis [36,37]. The transcription factor p53, encoded by the TP53 gene, can be detected in normal conditions but is silenced or mutated in cancer. DNA damage and oncogene activation can increase cellular p53 levels via phosphorylation and acetylation. p53 induces the extrinsic pathway and the mitochondrial pathway by regulating DRs and Bcl-2 proteins such as Bax and PUMA [38]. Cyclin-dependent kinases (CDKs) are regulators of cell-cycle progression and transcription. CDK transcriptionally enhances the activity of antiapoptotic BH3-only proteins such as myeloid cell leukemia (Mcl-1) while repressing sensitizers [39].

### 2.2. Mechanism of Necroptosis

As a counterpart of apoptosis, necrosis was long thought to be an unregulated form of cell death. However, in 2005, as suggested by Degterev et al. [40], necroptosis was reconsidered to be a necrotic form of PCD. The morphology of necroptotic cells is the same as that of necrotic cells, including a disrupted plasma membrane, and both types of cells passively release intracellular contents [41].

In necroptotic cells, death receptors (such as TNFR1 and Fas) and pattern recognition receptors (PRRs) such as Toll-like receptor 3 (TLR3) are activated by binding to their cognate ligands [15]. Then, these activated receptors recruit interacting kinase 1 (RIPK1) and a series of proteins to form an oligomeric complex, in which RIPK1 is polyubiquitinated and cleaved [42,43]. The NF-κB-dependent proinflammatory pathway and prosurvival pathway are involved in most of the abovementioned activities to promote cell survival [44]. An oligomeric complex, which consists of RIPK1, FADD, and CASP8, then exerts proapoptotic effects after dimerization by activating CASP8. However, in the absence of CASP8, active RIPK1 recruits and phosphorylates receptor-interacting kinase 3 (RIPK3) to form a RIPK1/RIPK3 complex, which then recruits and phosphorylates mixed lineage kinase domain-like (MLKL) to form the necrosome [42,45,46,47]. Phosphorylated MLKL can increase plasma membrane permeability by opening calcium or sodium ion channels or by directly forming pores in the plasma membrane, which leads to membrane rupture and the release of DAMPs. The released DAMPs inevitably cause inflammation and trigger immune responses [48,49].

### 2.3. Mechanism of Pyroptosis

Pyroptosis is an inflammatory form of PCD that was first observed in 1986 in primary mouse macrophages undergoing anthrax-induced lethality, which causes the rapid release of cell contents and cell death [50,51]. A series of caspase families are involved in pyroptosis in an inflammasome-dependent manner and are considered to play an indispensable role in the occurrence and development of cancer [51].

Pyroptosis is mediated through two main mechanisms: canonical and noncanonical pathways that trigger the gasdermin family [51,52]. In the canonical pathway, pathogen-associated molecular patterns (PAMPs) or DAMPs are detected by PRRs, such as TLRs and Nod-like receptors (NLRs), which activate inflammasome sensors, such as Nod-like receptor protein 3 (NLRP3), AIM2, and pyrin [51,53,54]. Activated inflammasome sensors recruit the adapter protein apoptosis-related speck-like protein (ASC) and pro-caspase-1 to form inflammasomes [55]. Procaspase-1 is then cleaved, which yields activated caspase-1, which can process procytokines such as interleukin (IL)-1β/18 to generate mature IL-1β/18. Activated IL-18/1β is secreted outside a cell through membrane pores and amplify the inflammatory response [56]. Activated caspase-1 can also process gasdermin D (GSDMD), thereby releasing the N-terminal fragment of GSDMD (GSDMD-N), which contributes to the nonselective formation of pores on the cell membrane and eventually results in cell swelling and lysis [57,58]. In the noncanonical pathway, gram-negative bacterial lipopolysaccharide (LPS) triggers pyroptosis by activating caspase-4/5 in humans or caspase-11 in mice to cleave GSDMD [58]. GSDMD-N positively regulates pyroptosis by activating the NLRP3 or NLRC4 inflammasome [59]. In addition to the abovementioned pyroptosis pathways, the caspase-3-mediated GSDME pathway has also been shown to play an essential role in inducing pyroptosis, especially during chemotherapeutic treatment [60].

### 2.4. The Connection between Apoptosis, Necroptosis, and Pyroptosis

The apoptosis, necroptosis, and pyroptosis pathways have long been considered to function in parallel but have recently been shown to be tightly connected and to interact with each other. Acting as a bridge between apoptotic and necroptotic pathways, caspase-8 is a well-known PCD mediator. It can not only activate the downstream executioner caspase-3/7 in the apoptosis pathway but also cleave RIPK1 and RIPK3 in the ripoptosome in the necroptosis pathway, thereby preventing necroptosis and facilitating apoptosis [61,62,63]. In addition, FADD and caspase-8 mediate caspase-1 processing, NLRP3 inflammasome assembly, and GSDMD activation, indicative of crosstalk between apoptosis and pyroptosis pathways [64,65,66]. In a recent discovery, pannexin-1, a channel-forming glycoprotein in macrophages, promoted NLRP3 inflammasome activation in the extrinsic and intrinsic apoptosis pathways [67]. Necroptosis can also play an important role in initiating pyroptosis. For example, RNA viruses upregulate the expression of NLRP3 inflammasome components in a RIPK1/RIPK3-dependent and MLKL-independent manner [68]. Furthermore, the PANoptosome, the set of activated apoptosis, necroptosis, and pyroptosis components that converge within the same period, was recently proposed and has received significant attention [69,70]. In PANoptosis, the connection between apoptosis, necroptosis, and pyroptosis is particularly tight, and the specific mechanisms will become clearer as relevant literature is published in the future.

## 3. Role of Apoptosis, Necroptosis, and Pyroptosis in PCa

### 3.1. Role of Apoptosis in Prostate Cancer

Previous studies have shown that the normal growth and function of the prostate gland as well as the growth and progression of PCa largely depend on signaling by androgen-induced AR pathway activation [71]. After treatment with androgen deprivation therapy or castration, prostate glands undergo marked involution characterized by apoptotic cell death, which significantly inhibits the progression of PCa. Furthermore, intrinsic pathways induced by mitochondrial or extrinsic death pathway receptors are both involved in the apoptosis of prostate epithelial cells after androgen deprivation therapy or castration [72].

The activation of death receptors initiates the extrinsic death pathway. Among the death receptors, TRAIL-R2 (DR5) is a member of the TNFR family, and drugs targeting DR5 are the most advanced [73]. TRAIL-R2 was downregulated in PCa cell lines and was markedly reduced in high-grade tumors [74]. The agonistic monoclonal antibody lexatumumab targeting DR5 has been the subject of early-phase investigations for use against several solid malignancies, including PCa [37]. The extrinsic death pathway can also be activated in immune responses. A recent study cocultured human NK cells (KHYG-1) with human prostate cancer stem-like cells. They discovered that NK cells exerted a killing effect by initiating the TRAIL/DR5 cell death pathway [75]. In another study, a selenium-bearing ruthenium complex (RuSe) was designed. RuSe potentiated NK cell-mediated killing of PC3 cells by activating TRAIL-R and FasL [76].

Bcl-2 family proteins and their regulators are also key molecules in PCa progression and therapy resistance. Initially, researchers discovered that the expression of the antiapoptotic protein Bcl-2 was decreased in PCa epithelial cells but increased after castration [77,78]. Moreover, increased levels of the antiapoptotic proteins Bcl-X and Mcl-1 were identified in prostate tumor cells, especially in high-grade and metastatic tumors [79,80]. Increased expression of antiapoptotic proteins contributes to PCa cell resistance to apoptosis mediated by androgen independence and metastasis [80,81]. The expression of the proapoptotic effector protein Bax was increased in castrated mice and was associated with poor outcomes, while the other Bcl-2 family member, Bak, was detected in PCa cell lines, and its level was increased in therapeutic assays [72]. In addition, BH3-only protein (such as BAD) levels were dysregulated in PCa cells, and this dysregulation was correlated with biochemical recurrence (BCR) and overall survival (OS) [82]. The aforementioned data indicate that the normal balance between antiapoptosis and proapoptosis pathway activation was disrupted in PCa, and as tumor progression increased, PCa cells gradually exhibited resistance to apoptosis. In addition, Bcl-2 family members can be regulated by transcription factors such as NF-κB and p53 and signaling pathways such as the PI3K/AKT and RAS/ERK signaling pathways. The direct or indirect regulation of the intrinsic pathway leads to significant benefits for the development of drugs that reduce PCa resistance to apoptosis [72,73].

### 3.2. Role of Necroptosis in Prostate Cancer

Necroptosis is considered a programmed form of necrosis characterized by mitochondrial alterations and plasma membrane permeabilization, which results in the release of cytoplasmic content into the extracellular space, leading to inflammatory reactions [52]. Accumulated evidence emphasizes the importance of necroptosis in PCa, and greater comprehension of the necroptotic mechanism might be helpful in creating novel strategies for controlling PCa [83].

A study performed bioinformatics analyses using a dataset from The Cancer Genome Atlas (TCGA) database and identified necroptosis-related genes that were closely associated with PCa prognosis. Using them as a gene signature to construct a prognostic model led to the accurate prediction of 1-, 3-, and 5-year OS in prostate adenocarcinoma (PRAD) patients [84]. For one study, researchers collected 67 prostate tissues that had been used to histologically diagnose PCa and categorized them into different tumor progression stages. RNA expression, tumor growth, etc., were measured. The results showed that the expression of RIPK3 was significantly elevated in the early stage but profoundly decreased in the final cancer stage [85]. This finding demonstrated that necroptosis was activated in the early stages of tumor progression but was resisted by tumor cells during disease progression into the late stage. Another study reported research on the biological role and clinical significance of RIP3 in the PCa context. They discovered that RIP3 was significantly downregulated in PCa cell lines and clinical prostate tumor samples. Upregulated RIP3 alleviates PCa progression by activating the RIP3/MLKL signaling pathway. Furthermore, RIP3 inhibits the proliferation and tumorigenicity of PCa cells in an MLKL-dependent manner both in vitro and in vivo [86]. These studies indicate that necroptosis is closely related to the progression of PCa and that inducing necroptosis in PCa cells is a feasible treatment strategy; however, a larger number of experimental and clinical studies are needed to confirm these conclusions and optimize the therapeutic approach.

### 3.3. Role of Pyroptosis in Prostate Cancer

Pyroptosis is an inflammatory form of PCD that can affect the tumor immune environment. The abnormal expression of pyroptosis-related genes (PRGs) may be closely related to the tumor immune microenvironment and thus promote the occurrence and development of PCa. By analyzing data collected in an online database, Hu and colleagues discovered that the expression levels of PRGs were significantly different between tumor and normal tissues. They constructed a prognostic signature based on these PRGs, their expression profile, a clinical database that can be used to precisely predict BCR after radical intervention, which is a determining risk factor for PCa specificity and distant metastasis, and the progression-free survival (PFS) rate [87]. Other researchers who repeated this work verified that the identified PRGs were closely related to carcinogenesis, tumor cell invasion, and the immune microenvironment in PCa [88,89].

Inflammation is an important prerequisite for the development of PCa [90]. The NLRP3 inflammasome is a well-studied inflammasome that induces pyroptosis in PCa. A recent study demonstrated that the expression level of NLRP3 was upregulated in LNCaP and PC3 cell lines. Activation of the NLRP3 inflammasome by LPS + ATP promoted the proliferation and migration of prostate tumor cell lines in vivo. NLRP3 knockdown inhibited the malignant progression of PCa cell lines [91]. Another study revealed the relationship between NLRP3 inflammasome and AR levels in high-grade PCa tumors. Compared with AR-dependent low-grade PCa cells, high-grade PCa cells showed increased NLRP3 levels. Furthermore, they identified an AR-related circular RNA (circRNA) named circAR-3-a, which can acetylate NLRP3 and promote NLRP3 inflammasome complex assembly. Disrupting NLRP3 acetylation or blocking inflammasome assembly with an inhibitor suppressed the progression of PCa xenograft tumors [92]. The NLRP12 inflammasome level was also shown to be significantly higher in malignant prostate tissues than in adjacent benign tissues, as indicated by immunostaining intensity [93].

Another pore-forming effector protein from the gasdermin family, GSDME, which is expressed in most normal tissue cells, is activated by apoptotic caspase (caspase 3), and its activation can switch the PCD from apoptosis to pyroptosis [94]. In contrast to NLRP3/caspase-1-mediated DSDMD cleavage, caspase-3-mediated DSDME cleavage plays a crucial role in the chemotherapeutic treatment of PCa. Tian et al. discovered that the expression level of GSDME in PCa cells did not significantly change, but its activity was profoundly increased by the poly polymerase (PARP) inhibitor olaparib. Simply upregulating GSDME conferred cells with sensitivity to olaparib but did not inhibit tumor cell proliferation [95]. These findings indicate that the DSDME level is physiologically low in both normal and PCa cells and that upregulation of DSDME may increase the sensitivity of PCa cells to chemotherapeutic drugs.

### 3.4. Role of PANoptosis in Tumors and PCa

PANoptosis is a recently proposed PCD that highlights the crosstalk and coordination among apoptosis, necroptosis, and pyroptosis pathways, which are interconnected via shared regulatory proteins and signaling pathways [96]. In combination with inflammatory cytokine-induced signaling through death domain-containing receptors, activated PRRs initiate this highly interconnected form of cell death, PANoptosis. PANoptosis has been linked to the development of multiple systemic diseases, including infectious diseases, cancers, neurodegenerative diseases, and inflammatory diseases [97]. An accumulation of recent studies performed bioinformatics analysis of expression data of PANoptosis-related genes (PRGs) based on online databases. They identified several PRGs correlated with patient survival, immune responses, and/or cancer-related biological processes. PRGs were used to construct PANoptosis signatures and significantly predicted the prognosis and immunotherapeutic response of several cancers, including pancreatic cancer, colon cancer, gastric cancer, and prostate adenocarcinoma (PRAD) [22,98,99,100]. However, in vivo and in vitro experiments on prostate cancer and PANoptosis have not been performed. These discoveries demonstrate the close connection between PANoptosis and tumors and lay the foundation for our deeper and more systematic research into the regulatory mechanisms underlying PANoptosis effects on PCa.

## 4. Therapeutic Potential of Compounds and Targets That Induce More than a Single Form of Apoptosis, Necroptosis, and Pyroptosis in PCa

A growing number of treatments have been found to induce more than just a single type of PCD. Compounds and targets previously thought to only induce apoptosis have been shown to have a positive effect on inducing pyroptosis and necroptosis. A series of newly discovered compounds and potential targets can induce two or even three forms of PCD in cancer. However, no compounds or targets have been found to induce PANoptosis (apoptosis, necroptosis, and pyroptosis) in PCa. Compounds and targets that induced both apoptosis and necroptosis or apoptosis and pyroptosis are listed in Table 1 and Table 2. Compounds and targets that induce PANoptosis in nonprostatic cancers are shown in Table 3.

### 4.1. Compounds and Targets That Induce Both Apoptosis and Necroptosis in PCa

Selenium, a fundamental and essential trace mineral for the biological activities of mammals, has been proven to be closely but negatively correlated with cancer risk and mortality [119]. A previous study showed that sodium selenite, an inorganic form of selenium compounds, sensitized LNCaP cells to TRAIL-induced apoptosis in a ROS/p53/Bax-mediated manner [101]. Recently, a study found that sodium selenite induced necrosis-like morphologic changes in PC-3 cells and DU145 cells, which is not caspase-mediated apoptosis, pyroptosis, or autophagic cell death. A RIP3 inhibitor protected PC-3 cells from selenite-induced cell death by inducing ATP depletion and inhibiting PFK activity [102]. Another study developed biogenic selenium nanoparticles, and oral administration of selenium nanoparticles induced markedly lower toxicity than the major natural foodform of selenium (L-selenomethionine) in mice. At a minimal concentration of 2 μg Se/mL, it could induce necroptosis in LNCaP-FGC cells through the upregulation of TNF and interferon regulatory factor (IRF1) [103].

The natural compound tocotrienol (TT) is a member of the vitamin E family and is expressed in four isoforms (α-TT, β-TT, γ-TT, and δ-TT), in which γ-TT and δ-TT specifically exert antitumoral activity, affecting proliferation, metastasis, and angiogenesis in different types of tumors [120]. Previous studies have proven that δ-tocotrienols (δ-TTs) exert antitumor effects by inducing apoptosis in PC3 and DU145 cells via triggering endoplasmic reticulum (ER) stress [104]. Another study revealed the involvement of the necroptosis pathway in the anticancer treatment of δ-TT in PCa cells (DU145 and PC3 cells). Inhibition of necroptosis prevents δ-TT-induced PCa cell death. They also evaluated the combined effect of δ-TT and docetaxel, and the combination was proven to be effective for PCa treatment. Furthermore, δ-TT induced death in DU145 cells that had developed docetaxel resistance by activating necroptosis [105]. In conclusion, the above study demonstrated that δ-TT shows the ability to induce necroptosis in the PC3, DU145, and DU-DXR cell lines, which indicates that the induction of necroptosis in PCa is a promising therapeutic strategy to overcome DTX chemoresistance.

Ophiopogonin D′ (OPD′) is a natural compound extracted from the traditional Chinese medicine Ophiopogon japonicus. Ophiopogonin D exerts potential anticancer effects by inducing cell cycle arrest and activating apoptosis and autophagy [121]. OPD′ was shown to exert potent antitumor activity against PC3 cells. It induced apoptosis via a RIPK1-related pathway, increased the protein expression levels of RIPK1 and Bim, and decreased the levels of cleaved-RIPK1, caspase 8, cleaved-caspase 8, Bid, caspase 10, and cleaved-caspase 10. OPD′ also increased the mRNA expression of Bim. The protein expression of Bim was decreased when cells were pretreated with necrostatin-1. Treatment with OPD′ inhibited the growth of PC3 and DU145 xenograft tumors in BALB/c nude mice [106]. A previous study indicated that OPD′ exhibited antitumor activity against AR-independent PCa [106]. In a recent study, OPD′ induced RIPK1-mediated necroptosis in AR-dependent LNCaP cells in a FasL-dependent manner. The RIPK1 inhibitor necrostatin-1 and the MLKL inhibitor necrosulfonamide both inhibited OPD′-activated necroptosis in LNCaP cells in a synergistic manner. Furthermore, OPD′ increased the Fas-associated necroptosis rate in LNCaP cells and regulated the expression levels of FasL, AR, and prostate-specific antigen in a RIPK1-dependent manner [107]. These results suggested that OPD′ may exhibit potential as an anti-PCa agent by inducing RIPK1- and MLKL-dependent necroptosis.

Shikonin, a natural compound extracted from the roots of *Lithospermum erythrorhizon*, has several protective effects, including anti-inflammatory and antitumor functions [122]. A study showed that exposure to SHI resulted in an accumulation of apoptotic events in parental and DX-resistant PC3 and DU145 cells. Although apoptosis was undetectable in parental LNCaP cells, a concentration of 0.7 µM SHI or higher contributed to apoptosis in the DX-resistant LNCaP cells. Furthermore, exposure to SHI showed an increase in pRIP1 and/or pRIP3 activation in PC3 and DU145 cells, which are indicative of necroptosis. However, SHI-induced necroptosis was more dominant than apoptosis in both parental and DX-resistant PCa cells for enhanced pRIP1 and pRIP3 expression and remarkable reversal of SHI’s antigrowth effect after applying the necroptosis inhibitor necrostatin-1 [108].

The sirtuin (SIRT) family of proteins, highly conserved NAD^+^-dependent histone deacetylases, are involved in several biological processes (such as oxidative stress responses, apoptosis, and inflammation), which makes SIRTs promising therapeutic targets for PCa [123]. A recent study found that SIRT6 was overexpressed in prostate tumors compared with normal or paratumor prostate tissues. Tissue microarray studies confirmed the higher levels of SIRT6 in both prostate tumor tissues and prostate cancer cells than in their normal counterparts. Knockdown of SIRT6 in human prostate cancer cells increased apoptosis by leading to sub-G1 phase arrest of the cell cycle and downregulating the expression of BCL2, as well as elevated DNA damage levels [109]. A study discovered that the expression levels of SIRT6 were significantly increased in PCa patients and were associated with the patients’ Gleason score and nodal metastasis number. SIRT6 promoted PCa progression by inhibiting RIPK3-mediated necroptosis and the innate immune response. Knockdown of SIRT6 not only activated TNF-induced necroptosis but also reestablished the corresponding recruitment of macrophages and neutrophils [110].

Reticulocalbin 1 (RCN1), an ER-resident Ca^2+^-binding protein, is essential for the accumulation of cell bioactivity. The dystegulation of RCN1 is correlated with cancer [124]. A recent study showed that reticulocalbin 1 is highly expressed in prostate cancer. RCN1 depletion led to the activation of caspase-3 and PARP in DU145 cells, which are markers of apoptosis. An inhibitor of necroptosis, necrostatin-1, evidently attenuated cell death in LNCaP cells treated with siRCN1, which suggested that necroptosis may partially contribute to siRCN1-mediated LNCaP cell death [111].

### 4.2. Compounds and Targets That Induce Apoptosis and Pyroptosis in PCa

Targeting pyroptosis therapeutic regulation induces pyroptosis mainly by inhibiting the formation of the NLRP3 inflammasome and DSDME-related pathway in PCa.

Ulinastatin (UTI), a serine protease inhibitor, is a glycoprotein composed of 143 amino acids. In humans, the precursor of UTI is an inactive interalpha-trypsin inhibitor synthesized by the liver. Ulinastatin exerts clear anti-inflammatory, antioxidant, and anti-apoptotic effects [125]. A recent experiment showed that UTI tended to suppress proliferation, migration, and invasion of cells but promoted apoptosis in PC-3 cells at higher concentrations. Moreover, UTI blocked RhoA and ROCK-mediated NLRP3 inflammasome activation in PC-3 cells [126].

Chalcones, which are flavones, possess two phenyl rings (the A- and B-rings) connected by a three-carbon α, β-unsaturated carbonyl bridge. A novel 3′,5′-diprenylated chalcone (C10) in the chalcone family exhibited anti-inflammatory and anticancer pharmacological functions in the context of leukemia. In one study, C10 profoundly inhibited the proliferation of human leukemia cell lines via its effect on the apoptosis and autophagy pathways [127]. In a recent study of the effects of C10 on PCa, C10 significantly reduced the proliferation and viability of PC3 cells by inducing caspase-dependent apoptosis and GSDME-dependent pyroptosis by activating PKCδ/JNK signaling [112].

Cell division cycle protein 20 (CDC20), the adaptor subunit and activator of the anaphase-promoting complex (APC), exerts clear effects on the cell cycle and tumorigenesis [128]. Previous findings indicated that oncogenic CDC20 was overexpressed in Pca, and knockdown of CDC20 resulted in inhibition of cell proliferation, migration, and tumor formation, induced cell apoptosis, and increased radiosensitivity in PCa in vitro and in vivo. Furthermore, CDC20 regulates the Twist1 pathway to influence cell proliferation and migration. These results suggest that targeting CDC20 and Twist1 to induce apoptosis may be an effective way to improve the radiosensitivity of PCa [129]. In one study, a set of PCa and mPCa data was downloaded from the Gene Expression Omnibus (GEO), and an integrated analysis was performed to identify differentially expressed genes. Higher expression of the CDC20 and PTTG1 genes was shown in mPCa samples and was significantly associated with a poorer prognosis. In addition, these activated genes promoted the migration of PCa cells [130]. Another study reported the discovery of CDC20 upregulation mediated by ubiquitination and the proteolysis of GSDME. Knockdown of CDC20 increased the level of GSDME by switching the PCD pathway from apoptosis to pyroptosis. Furthermore, depletion of CDC20 significantly enhanced antitumor immunity in a CD8+ T lymphocyte-dependent manner. Administration of a CDC20 inhibitor, such as Apcin, with α-PD-1 led to synergistic antitumor effects mediated via immunotherapy in murine models in vivo [113].

### 4.3. Compounds and Targets That Induce PANoptosis (Apoptosis, Necroptosis, and Pyroptosis) in Nonprostatic Cancers

Although PANoptosis has not been mentioned in PCa, a series of novel studies confirmed its existence in several nonprostatic cancers, which sets a precursor for research into PANoptosis in PCa. Cysteine desulfurase (NFS1) is an iron–sulfur (Fe–S) cluster biosynthetic enzyme. A study revealed that depletion of NFS1 significantly augmented the sensitivity of colorectal cancer (CRC) cells to oxaliplatin by triggering PANoptosis in a ROS-mediated manner in vitro and in vivo. Furthermore, a high expression level of NFS1 was correlated with poor survival and reduced sensitivity to chemotherapy in CRC patients [114]. Interferon regulatory factor 1 (IRF1) is a transcription factor induced in response to interferons. A study found that IRF1, which was downregulated in tumor tissues, prevents colitis-associated tumorigenesis in vivo in colorectal cancer. Mice with an IRF1 knockout showed attenuated PANoptosis cell death in the colon. However, the specific regulatory mechanism has not been further studied [115]. Cyclin-dependent kinase-1 (CDK1) is significantly related to the adverse clinical outcomes of adrenocortical carcinoma (ACC). A CDK1 inhibitor repressed the proliferation of ACC cells by triggering PANoptosis in a ZBP1-mediated manner [116]. Apurinic/apyrimidinic endonuclease 1 (APE1) is an enzyme that is broadly associated with a series of critical base excision repair pathways, including DNA repair, cancer cell growth, and drug resistance. A study revealed that APE1 is overexpressed in NSCLCs and correlates with malignancy. They identified an APE1 inhibitor named NO.0449-0145, which induced DNA damage, apoptosis, necroptosis, and pyroptosis in the NSCLC cell lines A549 and NCI-H460 [117]. Sulconazole has a broad spectrum of anticancer effects, including inhibiting the proliferation and migration of esophageal cancer cells. A study found that sulconazole induced not only PANoptosis but also ferroptosis by triggering mitochondrial oxidative stress and inhibiting glycolysis [118]. These studies suggest that PANoptosis may be a promising therapeutic target for anticancer therapy in cancers, including PCa.

## 5. Conclusions and Future Perspectives

Programmed cell death has attracted increasing attention from researchers globally and is a promising target for the treatment of PCa. In this review, apoptosis, necroptosis, pyroptosis, and PANoptosis, which are considered the main forms of PCD, play their own significant roles in the progression and therapeutic potential of PCa. Targets and compounds targeting not only apoptosis but also necroptosis, pyroptosis, and PANoptosis may effectively contribute to the development of new drugs to overcome the apoptosis resistance of PCa cells. However, compared with apoptosis, research on pyroptosis, necroptosis, and PANoptosis in PCa is still in the basic experimental stage. Therefore, in addition to improving their relevant mechanisms of action, clinical trials are needed to evaluate the feasibility of using agents targeting pyroptosis and necroptosis in clinical applications. Combining traditional Chinese medicine and its specific components with multidisciplinary approaches to identify drugs that can effectively induce PCD of cells in tumor sites may effectively promote the research and development of drugs related to PCa treatment. The efficiency and safety of drugs that induce apoptosis, necroptosis, pyroptosis, and PANoptosis in PCa urgently need to be evaluated. We hope that in the future, through cooperation among researchers in various disciplines and the development of biotechnology, feasible methods for targeting and precisely killing PCa cells will be proposed and put to use.

## Figures and Tables

**Figure 1 biomolecules-13-01715-f001:**
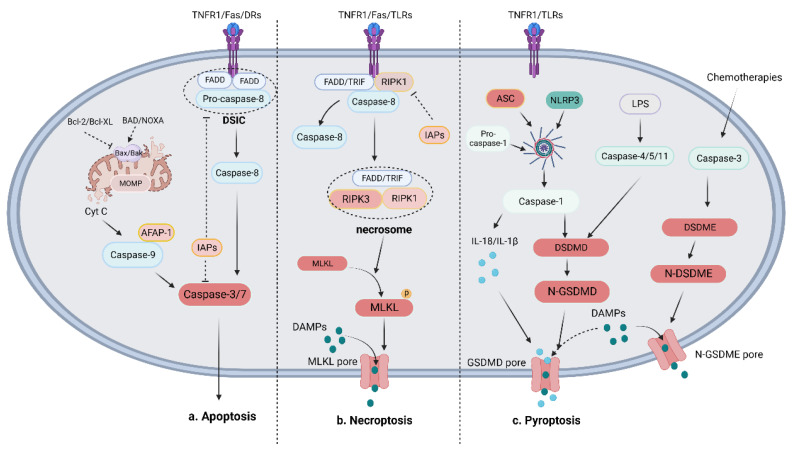
Mechanisms of apoptosis, necroptosis, and pyroptosis. (**a**) Mechanism of apoptosis: The apoptotic pathway is mainly initiated by the death receptor-mediated extrinsic pathway and the Bcl-mediated intrinsic pathway. The extrinsic pathway activates the effector caspase-3/7 to decompose cells via the formation of the DISC, which includes pro-caspase-8 and FADD, to cleave caspase-8. The intrinsic pathway is characterized by the release of mitochondrial Cyt C mediated by pro-apoptotic proteins (such as Bax and Bak) or anti-apoptotic proteins (such as Bcl-2, Bcl-XL, BAD, and NOXA). The combination of Cyt C, AFAP-1, and caspase-9 activates effector caspase-3/7 to crack cells. (**b**) Mechanism of necroptosis: The necroptotic pathway is initiated when caspase-8 is absent by RIPK1-mediated RIPK3 activation. MLKL is then recruited and phosphorylated to form porepores in the membrane and the release of DAMPs. (The circle P attached on MLKL is phosphate group). (**c**) Mechanism of pyroptosis: The pyroptotic pathway can be divided into the NLRP3-induced canonical pathway and the LPS-induced noncanonical pathway, which activates caspase-1 and GSDMD to cause the release of IL-18, IL-1β, and DAMPs. In addition to the above two main pathways, chemotherapies induce caspase-3-mediated GSDME cleavage to cleave cells and release DAMPs.

**Table 1 biomolecules-13-01715-t001:** Compounds and targets that induce both apoptosis and necroptosis in PCa.

Compound/Target	Induced Cell Death	In Vivo/Vitro	Cell Lines/Animals	Mechanisms	References
Sodium Selenite	Apoptosis	In Vitro	LNCaP Cells	Sensitized LNCaP Cells to TRAIL in a ROS/p53/Bax-mediated manner	[101]
	Necroptosis	In Vitro	PC-3, DU145 Cells	Induced RIP3/MLKL-independent necroptosis	[102]
Biogenic Selenium Nanoparticles	Necroptosis	In Vitro and In Vivo	LNCaP-FGC Cells/Mice	Upregulated the expression of TNF and IRF1	[103]
δ-TT	Apoptosis	In Vitro	DU145, PC3 Cells	Triggers endoplasmic reticulum (ER) stress	[104]
	Necroptosis	In Vitro	DU145, PC3 and DU-DXR Cells	Activation of the RIP3/MLKL pathway	[105]
Ophiopogonin D′	Apoptosis	In Vitro and In Vivo	PC3, DU145 Cells/Mice	Increased the expression of RIPK1 and Bim	[106]
	Necroptosis	In Vitro	LNCaP Cells	Induced the RIPK1/MLKL-mediated necroptosis in a FasL-dependent manner.	[107]
Shikonin	Apoptosis	In Vitro	Parental and DXR PC3 and DU145 Cells, DXR LNCaP Cells	Increased the expression of PARP, caspase 3, and caspase 8	[108]
	Necroptosis	In Vitro	PC3, DU145 Cells	Increased the expression of pRIP1 and/or pRIP3	[108]
SIRT6	Apoptosis	In Vitro and In Vivo	PC3 Cells/Mice	Elevated DNA damage level and decreased BCL2 gene expression	[109]
	Necroptosis	In Vitro	LNCaP Cells	Inhibited the expression of RIPK3	[110]
Reticulocalbin 1	Apoptosis	In Vitro and In Vivo	DU145 Cells/Mice	Activated caspase-3, PARP and ER stress	[111]
	Necroptosis	In Vitro	LNCaP Cells	Necrostatin-1 (inhibitor of necroptosis) reversed the cell death in siRCN1-treated LNCaP cells	[111]

**Table 2 biomolecules-13-01715-t002:** Compounds and targets that induce both apoptosis and pyroptosis in PCa.

Compound/Target	Cell Death	In Vivo/Vitro	Cell Lines/Animals	Mechanisms	References
Ulinastatin	Apoptosis	In Vitro	PC-3 Cells	Downregulation of anti-apoptotic protein Bcl-2 and upregulation of pro-apoptotic proteins (Bax, caspase3, and caspase-9)	[101]
	Pyroptosis	In Vitro	PC-3 Cells	Blocked the activation of NLRP3 inflammasome	[101]
3′,5′-diprenylated chalcone	Apoptosis	In Vitro	PC3 Cells	Cleavage of PARP, Caspase-3, Caspase-8, Caspase-9, Bax, and cytochrome C	[112]
	Pyroptosis	In Vitro	PC-3 Cells	Activation of GSDME	[112]
CDC20	Apoptosis	In Vitro and In Vivo	PC3, DU145 Cells/Mice	Increased the expression of RIPK1 and BIM	[106]
	Pyroptosis	In Vitro	Prostate Cancer Stem-like Cells	Downregulation of CDC20 increased the level of GSDME by transforming apoptosis to pyroptosis	[113]

**Table 3 biomolecules-13-01715-t003:** Compounds and targets that induce three of PANoptosis (apoptosis, necroptosis, and pyroptosis) in nonprostatic cancers.

Compound/Target	Cancers	In Vivo/Vitro	Cell Lines/Animals	Mechanisms	References
NFS1	Colorectal Cancer	In Vitro and In Vivo	Colorectal Cancer Cells/Mice	Increased the intracellular levels of ROS.	[114]
IRF1	Colorectal Cancer	In Vitro and In Vivo	Bone marrow–derived macrophages/Mice	Knockout of IRF1 attenuated PANoptosis	[115]
CDK1	Adrenocortical Carcinoma	In Vitro and In Vivo	SW-13 cells, NCI-H295R Cells/Mice	Binded with the PANoptosome in a ZBP1-dependent way	[116]
APE1	Non-Small Cell Lung Cancer	In Vitro	A549, NCI-460 Cells	Inhibited APE1 induced DNA damage and PANoptosis	[117]
Sulconazole	Esophageal Cancer	In Vitro	KYSE30, KYSE150 Cells	Triggered oxidative stress and inhibited glycolysis	[118]

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
