# Peer review of "Caspase-Linked Programmed Cell Death in Prostate Cancer: From Apoptosis, Necroptosis, and Pyroptosis to PANoptosis"

_biomolecules, 2023, doi:10.3390/biom13121715_

Round 1

Reviewer 1 Report

Comments and Suggestions for Authors

This review does not provide any additional information compared with very recent work and/or reviews already published on the subject:

Regulated Cell Death in Urinary Malignancies. Nie Z, Chen M, Gao Y, Huang D, Cao H, Peng Y, Guo N, Zhang S. Front Cell Dev Biol. 2021 Nov 12;9:789004. doi: 10.3389/fcell.2021.789004.

Relationship between pyroptosis-mediated inflammation and the pathogenesis of prostate disease. Zhao M, Guo J, Gao QH, Wang H, Wang F, Wang ZR, Liu SJ, Deng YJ, Zhao ZW, Zhang YY, Yu WX. Front Med (Lausanne). 2023 Jan 19;10:1084129. doi: 10.3389/fmed.2023.1084129.

Pyroptosis in urinary malignancies: a literature review. Wang S, Liao X, Xiong X, Feng D, Zhu W, Zheng B, Li Y, Yang L, Wei Q. Discov Oncol. 2023 Jan 26;14(1):12. doi: 10.1007/s12672-023-00620-7.

Construction of PANoptosis signature: Novel target discovery for prostate cancer immunotherapy. Yi X, Li J, Zheng X, Xu H, Liao D, Zhang T, Wei Q, Li H, Peng J, Ai J. Mol Ther Nucleic Acids. 2023 Jul 15;33:376-390. doi: 10.1016/j.omtn.2023.07.010.

In addition, the authors review the regulation of cell death (apoptosis, necroptosis, pyroptosis), which is already well known and described, and their connections.

Reviewer 2 Report

Comments and Suggestions for Authors

The authors Minggang Zhu, Di Liu, Guoqiang Liu, Mingrui Zhang, and Feng Pan present a nice coverage of programmed cell death involving caspases at some level. I must say that I enjoyed reading the review. It is a well-written compilation and runs very smoothly. I have a few minor comments about the review.

1.     The review title is “programmed cell death in Prostate Cancer; however, it excludes an important cell death mechanism, ferroptosis. I understand ferroptosis may not fit due to certain reasons or due to the focus of the review; I suggest using a term that is directly indicative of Programmed cell death except ferroptosis. I have a few suggestions, such as caspase-linked programmed cell death or caspase-mediated cell death. However, I leave it open to authors if they come up with other alternatives.

2.     I also suggest that the authors mention the criteria about what type of cell death mechanisms are included and what was excluded in the review.

3.     In many places, there were issues with the references shown as errors, e.g., lines 73, 312-314. I believe these occurred due to some issues with the citation manager.

4.     Text about the role of ∂-TT in PCa (lines 339-343) requires citation.

5.     Line 383, knockdown of SIRT6 in human prostate cancer cells led to sub-G1 phase arrest… Cells in subG1 are hypodiploid and are often dead. I suggest it should be corrected.

6.     In line 418, the authors state their own work as per the statement, Please check if it was cited appropriately.

Round 2

Reviewer 1 Report

Comments and Suggestions for Authors

I stand by my initial assessment. I should point out, however, that I had read the publications I quoted, so I didn't need you to remind me of their content.